# SimPLR: A Simple and Plain Transformer for Efficient Object Detection and Segmentation

**Duy-Kien Nguyen**                                        *d.k.nguyen@uva.nl*
*University of Amsterdam*

**Martin R. Oswald**                                        *m.r.oswald@uva.nl*
*University of Amsterdam*

**Cees G. M. Snoek**                                        *cgmsnoek@uva.nl*
*University of Amsterdam*

**Reviewed on OpenReview:** *https://openreview.net/forum?id=6LO1y8ZE0F*

## Abstract

The ability to detect objects in images at varying scales has played a pivotal role in the design of modern object detectors. Despite considerable progress in removing hand-crafted components and simplifying the architecture with transformers, multi-scale feature maps and pyramid designs remain a key factor for their empirical success. In this paper, we show that shifting the multiscale inductive bias into the attention mechanism can work well, resulting in a plain detector 'SimPLR' whose backbone *and* detection head are both non-hierarchical and operate on single-scale features. We find through our experiments that SimPLR with scale-aware attention is plain and simple architecture, yet competitive with multi-scale vision transformer alternatives. Compared to the multi-scale and single-scale state-of-the-art, our model scales better with bigger capacity (self-supervised) models and more pre-training data, allowing us to report a consistently better accuracy and faster runtime for object detection, instance segmentation, as well as panoptic segmentation. Code is released at https://github.com/kienduynguyen/SimPLR.

## 1 Introduction

After its astonishing achievements in natural language processing, the transformer (Vaswani et al., 2017) has quickly become the neural network architecture of choice in computer vision, as evidenced by recent success in image classification (Liu et al., 2021; Dosovitskiy et al., 2021; Dehghani et al., 2023), object detection (Carion et al., 2020; Zhu et al., 2021; Nguyen et al., 2022; Lin et al., 2023) and segmentation (Wang et al., 2021a; Cheng et al., 2022; Li et al., 2023). Unlike natural language processing, where the same pre-trained network can be deployed for a wide range of downstream tasks with only minor modifications (Brown et al., 2020; Devlin et al., 2019), computer vision tasks such as object detection and segmentation require a different set of domain-specific knowledge to be incorporated into the network. Consequently, it is commonly accepted that a modern object detector contains two main components: a pre-trained backbone as the *general* feature extractor, and a *task-specific* head that conducts detection and segmentation tasks using domain knowledge. For transformer-based vision architectures, the question remains whether to add more inductive biases or to learn them from data.

The spatial nature of image data lies at the core of computer vision. Besides learning long range feature dependencies, the ability of capturing local structure of neighboring pixels is critical for representing and understanding the image content. Building upon the successes of convolutional neural networks, a line of research biases the transformer architecture to be *multi-scale* and *hierarchical* when dealing with the image input, *i.e.*, Swin Transformer (Liu et al., 2021) and others (Fan et al., 2021; Wang et al., 2021b; Heo et al.,

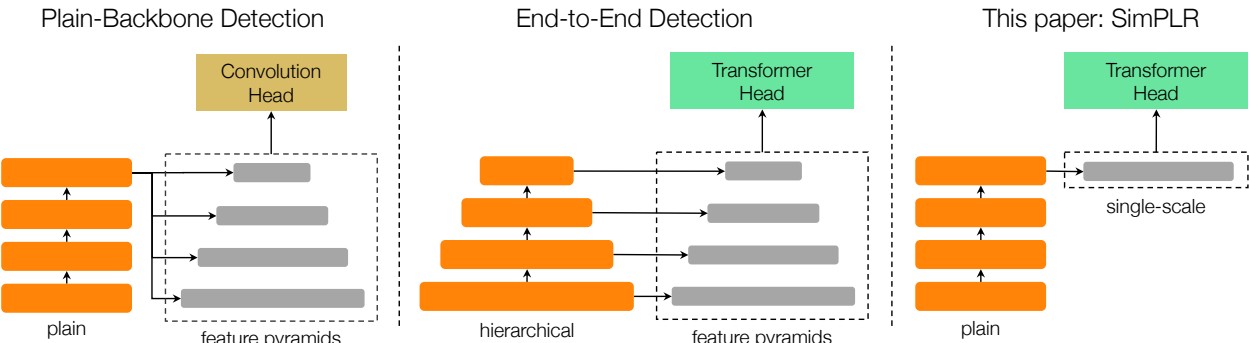

Figure 1: **Object detection architectures. Left:** The plain-backbone detector from (Li et al., 2022) whose input (denoted in the dashed region) are multi-scale features. **Middle:** State-of-the-art end-to-end detectors (Nguyen et al., 2022; Li et al., 2023; Hu et al., 2023) utilize a hierarchical backbone (*i.e.*, Swin (Liu et al., 2021)) to create multi-scale inputs. **Right:** Our simple single-scale detector following the end-to-end framework. Where existing detectors require feature pyramids to be effective, we propose a plain detector, SimPLR, whose backbone and detection head are non-hierarchical and operate on a single-scale feature map. The plain detector, SimPLR, performs on par or even better compared to hierarchical and/or multi-scale counterparts, while also being more scaling-efficient.

2021). The hierarchical design makes it easy to create multi-scale features for dense vision tasks and allows pre-trained transformers to be seamlessly integrated into a convolution-based detection head with a feature pyramid network (Lin et al., 2017), yielding impressive results in object detection and segmentation. However, the inductive biases in the architectural design make it benefit less from self-supervised learning and the scaling of model size (Li et al., 2022).

An alternative direction pursues the idea of a simple transformer with "less inductive biases" and emphasizes learning vision-specific knowledge directly from image data. Specifically, the Vision Transformer (ViT) (Dosovitskiy et al., 2021) stands out as a *plain* architecture with a constant feature resolution, and acts as the feature extractor in plain-backbone detection. This is motivated by the success of ViTs scaling behaviours in visual recognition (He et al., 2022; Bao et al., 2022; Dehghani et al., 2023). In addition, the end-to-end detection framework proposed by Carion et al. (2020) with a transformer-based detection head further removes many hand-designed components, like non-maximum suppression or intersection-over-union computation, that encodes the prior knowledge for object detection.

The plain design of ViTs, however, casts doubts about its ability to capture information of objects across multiple scales. While recent studies (Dosovitskiy et al., 2021; Li et al., 2022) suggest that ViTs with global self-attention could potentially learn translation-equivariant and scale-equivariant properties during training, leading object detectors still require multi-scale feature maps and/or an hierarchical backbone. This observation holds true for both convolutional (Ren et al., 2015; He et al., 2017; Li et al., 2022) and transformer-based detectors (Nguyen et al., 2022; Cheng et al., 2022; Lin et al., 2023; Li et al., 2023). Unlike hierarchical backbones, the creation of feature pyramids conflicts with the original design philosophy of ViTs. Therefore, our goal is to pursue a plain detector whose backbone *and* detection head are both *single-scale* and *non-hierarchical*. This is accomplished by directly incorporating multi-scale information into the attention computation.

In this paper, we introduce SimPLR, a plain detector for *both* end-to-end detection and segmentation frameworks. In particular, our detector extracts the single-scale feature map from a ViT backbone, which is then fed into the transformer encoder-decoder via a simple projection to make the prediction (see Fig. 1). To deal with objects of various sizes, we propose to incorporate scale information into the attention mechanism, resulting in an *adaptive-scale* attention. The proposed attention mechanism learns to capture adaptive scale distribution from training data. This eliminates the need for multi-scale feature maps from the ViT backbone, yielding a simple and efficient detector.

The proposed detector, SimPLR, turns out to be an effective solution for plain detection and segmentation. We find that multi-scale feature maps are not necessary and the scale-aware attention mechanism, that learns multi-scale information in the attention computation, adequately captures objects at various sizes from output features of a ViT backbone. Despite the plain architecture, our detector shows competitive performance compared to the strong hierarchical-backbone or multi-scale detectors, while being consistently faster. Moreover, the effectiveness of our detector is observed not only in object detection but also in instance segmentation and panoptic segmentation. Finally, SimPLR shows to be a scaling-efficient detector when moving to bigger models, indicating plain detectors to be a promising direction for dense prediction tasks.

## 2 Related Work

**Backbones for object detection.** Inspired by R-CNN (Girshick et al., 2014), modern object detection methods utilize a task-specific head on top of a pre-trained backbone. Initially, object detectors (Simonyan & Zisserman, 2015; Xie et al., 2017; He et al., 2016; Huang et al., 2017) were dominated by convolutional neural network (CNN) backbones (LeCun & Bengio, 1995) pre-trained on ImageNet (Deng et al., 2009). With the success of the transformer in learning from large-scale text data (Brown et al., 2020; Devlin et al., 2019), many studies have explored the transformer for computer vision (Chen et al., 2020a; Dosovitskiy et al., 2021; Liu et al., 2021). The Vision Transformer (ViT) (Dosovitskiy et al., 2021) with a simple design demonstrated the capability of learning meaningful representation for visual recognition. By removing the need for labels, methods with self-supervised learning have emerged as an even more powerful solution for pre-training general vision representations (Chen et al., 2020b; He et al., 2022). We show through experiments that SimPLR takes advantages of the significant progress in representation learning at scale with ViTs for object detection and segmentation.

**End-to-end detection and segmentation.** The end-to-end framework for object detection proposed in DETR (Carion et al., 2020) removes the need for many hand-crafted modules. This was made possible by adopting the Transformer as the detection head to directly give the prediction. Follow-up works (Dong et al., 2021; Wang et al., 2021a; Nguyen et al., 2022) extended the transformer-based head in end-to-end frameworks for instance segmentation, and panoptic segmentation. This inspired MaskFormer (Cheng et al., 2021) and K-Net (Zhang et al., 2021) to unify segmentation tasks with a class-agnostic mask prediction. Pointing out that MaskFormer and K-Net lag behind specialized architectures, Cheng *et al.* (Cheng et al., 2022) introduce Mask2Former, reaching strong performance on segmentation tasks. Yu *et al.* (Yu et al., 2022) replace self-attention with $k$-means clustering, further boosting the effectiveness of the network. Another direction is to improve the object query in the decoder via a denoising process (Zhang et al., 2023; Li et al., 2023) or a matching process (Jia et al., 2023; Zong et al., 2023). While simplifying the detection and segmentation framework, these architectures still require an hierarchical backbone along with feature pyramids. The use of feature pyramids increases the sequence length of the input to the transformer-based detection head, making the detector less efficient. In this work, we enable a plain detector by removing hierarchical and multi-scale constraints.

**Plain detectors.** Following the goal of less inductive biases in the architecture, recent studies focus on the non-hierarchical and single-scale detector. Motivated by the success of ViT, a line of research considers plain-backbone detectors that replace the hierarchical backbone with a ViT. Initially, Chen *et al.* (Chen et al., 2022) present UViT as a plain detector that contains a ViT backbone and a single-scale convolutional detection head. Since the backbone architecture is modified *during pre-training* to adopt the progressive window attention, UViT is unable to take the advantages of existing pre-training approaches with ViTs. ViTDet (Li et al., 2022) tackles this problem with simple adaptations of the ViT backbone *during fine-tuning*. These simple modifications allow ViTDet to benefit directly from recent self-supervised learning with ViTs (*i.e.*, MAE (He et al., 2022)), resulting in strong results when scaling to larger models. Despite enabling plain-backbone detectors, feature pyramids are still an important factor in ViTDet to detect objects at various scales.

Most recently, Lin *et al.* (Lin et al., 2023) introduce the transformer-based detector, PlainDETR, which also removes the multi-scale input. However, it still relies on multi-scale features to generate the object proposals for its decoder. In the decoder, PlainDETR replaces the standard Hungarian matching (Kuhn, 1955) with

hybrid matching (Jia et al., 2023) to strengthen its prediction, while our decoder preserves a simple design as in Zhu et al. (2021); Nguyen et al. (2022). Moreover, PlainDETR can only perform object detection while SimPLR facilitates detection, instance segmentation and panoptic segmentation. We believe to be the first to remove the hierarchical and multi-scale constraints which appear in the backbone *and* the input of the transformer encoder for *both* detection and segmentation tasks. Our proposed scale-aware attention can further plug into current end-to-end frameworks without significant architectural changes. Because of the simple and plain design, SimPLR is more efficient and effective when scaling to larger models.

## 3 SimPLR: A Simple and Plain Detector

Multi-scale feature maps in a hierarchical backbone can be easily extracted from the pyramid structure (Liu et al., 2016; Lin et al., 2017; Zhu et al., 2021). When moving to a ViT backbone with a constant feature resolution, the creation of multi-scale feature maps requires complex backbone adaptations. Moreover, the benefits of multi-scale features in object detection frameworks using ViTs remain unclear. Studies on plain-backbone detection (Li et al., 2022; Chen et al., 2022) conjecture the high-dimensional ViT with self-attention and positional embeddings (Vaswani et al., 2017) is able to preserve important information for localizing objects. From this conjecture, we hypothesize that a proper design of the transformer-based head will enable a plain detector.

Our proposed detector, SimPLR, is conceptually simple: a pre-trained ViT backbone to extract plain features from an image, which are then fed into a single-scale encoder-decoder to make the final prediction. Thus, SimPLR eliminates the non-trivial creation of feature pyramids from the ViT backbone. But the single-scale encoder-decoder requires an effective design to deal with objects at different scales. First, we review box-attention in Nguyen et al. (2022) as our strong baseline in end-to-end detection and segmentation using feature pyramids. Then, we introduce the key elements of our plain detector, SimPLR, including its *scale-aware* attention that is the main factor for learning of adaptive object scales.

### 3.1 Background

Our goal is to further simplify the detection and segmentation pipeline from Zhu et al. (2021); Li et al. (2022); Nguyen et al. (2022), and to prove the effectiveness of the plain detector in *both* object detection and segmentation tasks. We utilize the sparse box-attention mechanism from Nguyen et al. (2022) as strong baseline due to its effectiveness in learning discriminative object representations while being lightweight in computation.

We first revisit box-attention proposed by Nguyen *et al.* Nguyen et al. (2022). Given the input feature map from the backbone, the encoder layers with box-attention will output contextual representations. The contextual representations are utilized to predict object proposals and to initialize object queries for the decoder. We denote the input feature map of an encoder layer as $e \in \mathbb{R}^{H_e \times W_e \times d}$ and the query vector $q \in \mathbb{R}^d$, with $H_e, W_e, d$ denoting height, width, and dimension of the input features respectively. Each query vector $q \in \mathbb{R}^d$ in the input feature map is assigned a reference window $r=[x,y,w,h]$, where $x,y$ indicate the query coordinate and $w,h$ are the size of the reference window both being normalized by the image size. The box-attention refines the reference window into a region of interest, $r'$, as:

$$r' = F_{\text{scale}}\big(F_{\text{translate}}(r,q),q\big) \ , \tag{1}$$

$$F_{\text{scale}}(r,q) = [x, y, w + \Delta_w, h + \Delta_h] \ , \tag{2}$$

$$F_{\text{translate}}(r,q) = [x + \Delta_x, y + \Delta_y, w, h] \ , \tag{3}$$

where $F_{\text{scale}}$ and $F_{\text{translate}}$ are the scaling and translation transformations, $\Delta_x, \Delta_y, \Delta_w$ and $\Delta_h$ are the offsets regarding to the reference window $r$. A linear projection ($\mathbb{R}^d \to \mathbb{R}^4$) is applied on $q$ to predict offset parameters (*i.e.*, $\Delta_x, \Delta_y, \Delta_w$ and $\Delta_h$) w.r.t. the window size.

Similar to self-attention Vaswani et al. (2017), box-attention aggregates $n$ multi-head features from regions of interest:

$$\text{MultiHeadAttention} = \text{Concat}(\text{head}_1, \ldots, \text{head}_n)\, W^O \ , \tag{4}$$

During the $i$-th attention head computation, a $2\times2$ feature grid is sampled from the corresponding region of interest $r_i'$, resulting in a set of value features $v_i \in \mathbb{R}^{2\times2\times d_h}$. The $2\times2$ attention scores are efficiently generated by computing a dot-product between $q \in \mathbb{R}^d$ and relative position embeddings ($k_i \in \mathbb{R}^{2\times2\times d}$) followed by a softmax function. The attended feature $\text{head}_i \in \mathbb{R}^{d_h}$ is a weighted average of the $2\times2$ value features in $v_i$ with the corresponding attention weights:

$$\alpha = \text{softmax}(q^\top k_i) \ , \tag{5}$$

$$\text{head}_i = \text{BoxAttention}(q, k_i, v_i) = \sum_{j=0}^{2\times2} \alpha_j v_{i_j} \ , \tag{6}$$

where $q \in \mathbb{R}^d$, $k_i \in \mathbb{R}^{2\times2\times d}$, $v_i \in \mathbb{R}^{2\times2\times d_h}$ are query, key and value vectors of box-attention, $\alpha_j$ is the $j$-th attention weight, and $v_{i_j}$ is the $j$-th feature vector in the feature grid $v_i$. To better capture objects at different scales, the box-attention in Nguyen et al. (2022) takes $s$ multi-scale feature maps, $\{e^j\}_{j=1}^s (s = 4)$, as its inputs. In the $i$-th attention, $s$ feature grids are sampled from each of multi-scale feature maps in order to produce $\text{head}_i$.

Sparse attention mechanisms like box-attention lie at the core of recent end-to-end detection and segmentation models due to their ability of capturing object information with lower complexity. The effectiveness and efficiency of these attention mechanisms bring up the question: *Is multi-scale object information learnable within a detector that is non-hierarchical and single-scale?*

## 3.2 Scale-aware attention

The output features of the encoder should capture objects at different scales. Therefore, unlike the feature pyramids where each set of features encodes a specific scale, predicting objects from a plain feature map requires its feature vectors to reason about dynamic scale information based on the image content. This can be addressed effectively by a multi-head attention mechanism that capture different scale information in each of its attention heads. In that case, global self-attention is a potential candidate because of its large receptive field and powerful representation. However, its computational complexity is quadratic w.r.t. the sequence length of the input, making the operation computationally expensive when dealing with high-resolution images. The self-attention also leads to worse performance and slow convergence in end-to-end detection (Zhu et al., 2021). This motivates us to propose a multi-head *scale-aware* attention mechanism for single-scale input.

**Scale-aware attention.** In sparse multi-scale attention mechanisms, such as deformable attention (Zhu et al., 2021) or box-attention (Nguyen et al., 2022), each feature vector is assigned to a scale in the feature pyramid. As a result, feature vectors learn to adapt to that specific scale. While this behaviour may not impact the multi-scale deformable attention or multi-scale box-attention – which utilizes feature pyramids for detecting objects – it poses a big challenge in learning scale-equivariant features from a single-scale input.

To address this limitation, we propose two variants of multi-head *scale-aware* attention (*i.e.*, *fixed-scale* and *adaptive-scale*) that integrate different scales into each attention head, allowing query vectors to choose the suitable scale information during training. Our proposed attention mechanism is simple: we assign anchors of $m$ different scales to attention heads of each query. We use $m$ anchors with size $w{=}h \in \{s \cdot 2^j\}_{j=0}^{m-1}$, where $s$ is the size of the smallest anchor, and $m$ is the number of scales. Surprisingly, our experiments show that the results are *not* sensitive to the size of the anchor, as long as *a sufficient number of scales* is used.

i) *Fixed-Scale Attention.* Given anchors of $m$ scales, we distribute them to $n$ attention heads in a round-robin manner. Thus, in multi-head fixed-scale attention, $\frac{n}{m}$ attention heads are allocated for each of the anchor scales. This uniform distribution of different scales enables fixed-scale attention to learn diverse information from local to more global context. The aggregation of $n$ heads results in scale-aware features, that is suitable for predicting objects of different sizes.

ii) *Adaptive-Scale Attention.* Instead of uniformly assigning $m$ scales to $n$ attention heads, the adaptive-scale attention learns to allocate a scale distribution based on the context of the query vector. This comes from the motivation that the query vector belonging to a small object should use more attention heads for capturing fine-grained details rather than global context, and vice versa.

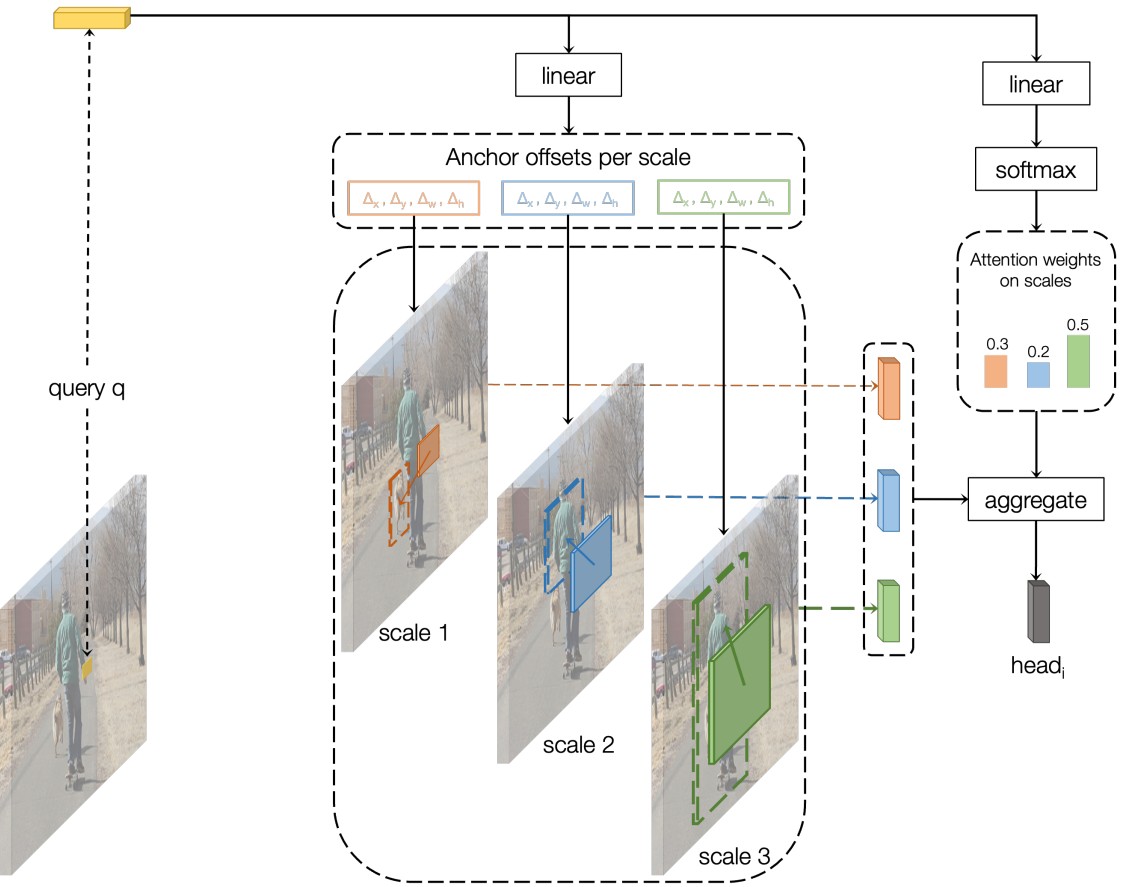

Figure 2: **Illustration of the proposed adaptive-scale attention.** In the $i$-th attention head, given a query vector and anchors of three scales, the adaptive-scale attention learns to attend to a region of interest w.r.t. each scale. It then generates attention weights on these scales adaptively based on the query vector to produce head$_i$. As this mechanism allows each vector in our plain feature map to learn suitable scale information during the training process, we no longer need an hierarchical backbone along with feature pyramids.

In each attention head, given the query vector $q \in \mathbb{R}^d$ in the input feature map and $m$ anchors of different scales, $\{r_j\}_{j=0}^{m-1}$, the adaptive-scale attention predicts offsets of all anchors, $\{\Delta_{x_j}, \Delta_{y_j}, \Delta_{w_j}, \Delta_{h_j}\}_{j=0}^{m-1}$. Besides, we apply a scale temperature to each set of offsets before the transformations:

$$F_{\text{scale}}(r_j, q) = \left[ x, y, w + \Delta_w \cdot \frac{2^j}{\lambda}, h + \Delta_h \cdot \frac{2^j}{\lambda} \right], \tag{7}$$

$$F_{\text{translate}}(r_j, q) = \left[ x + \Delta_x \cdot \frac{2^j}{\lambda}, y + \Delta_y \cdot \frac{2^j}{\lambda}, w, h \right], \tag{8}$$

where $\frac{2^j}{\lambda}$ is the scale temperature corresponding to $r_j$. The scale temperature allows the transformation functions to capture regions of interest, $r'_j$, corresponding to the scale of anchors. The adaptive-scale attention produces a feature vector per scale from the corresponding region of interest. The attention weights on scales are then generated by applying a linear projection on query vector $q$ followed by softmax normalization. The attended feature, head$_i$, is the weighted sum of feature vectors with the corresponding attention scores of $m$ scales (see Fig. 2). This allows the attention mechanism to learn suitable scale distribution based on the content of query vector. The adaptive-scale attention provides efficiency due to sparse sampling and strong flexibility to control scale distribution via its attention computation.

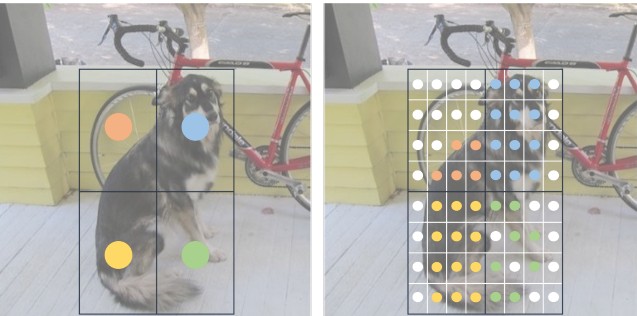

Figure 3: **Masked Instance-Attention. Left:** Box-attention (Nguyen et al., 2022) samples $2 \times 2$ grid features in the region of interest. **Right:** Proposed masked instance-attention for dense grid sampling. The $2 \times 2$ attention scores are denoted in four colours and the masked attention scores are in white. The masked instance-attention preserves the efficiency by sparse sampling while better capturing objects of different shapes.

### 3.3 Object representation for panoptic segmentation

Panoptic segmentation proposed by Kirillov et al. (2019) requires the network to segment both "things" and "stuff". To enable the plain detector on panoptic segmentation, we make an adaptation in the mask prediction of SimPLR. Following Cheng et al. (2022), we predict segmentation masks of both types by computing the dot-product between object queries and a high-resolution feature map (*i.e.*, $\frac{1}{4}$ feature scale). As the ViT and SimPLR encoder features are of lower resolution, we simply interpolate the last encoder layer to $\frac{1}{4}$ scale and stack $K{=}1$ scale-aware attention layer on top. This simple modification produces a high resolution feature map that is beneficial for learning fine-grained details.

**Masked instance-attention.** In order to learn better object representation in the decoder, we propose masked instance-attention that is also a sparse attention mechanism for efficiency. The masked instance-attention follows the grid sampling strategy of the box-attention in Nguyen et al. (2022), but improves the attention computation to better capture objects of different shapes.

To be specific, the region of interest $r'$ is divided into 4 bins of $2 \times 2$ grid, each of which contains a $\frac{m}{2} \times \frac{m}{2}$ grid features sampled using bilinear interpolation. Instead of assigning an attention weight to each feature vector, a linear projection ($\mathbb{R}^d \to \mathbb{R}^{2 \times 2}$) is adopted to generate the $2 \times 2$ attention scores for 4 bins. The $\frac{m}{2} \times \frac{m}{2}$ feature vectors within the same bin share the same attention weight. Inspired by Cheng et al. (2022), we utilize the mask prediction of the previous decoder layer as the attention mask to the attention scores (see Fig. 3).

$$A = \text{repeat}(\alpha) \tag{9}$$
$$\text{head}_i = \text{softmax}(A + \mathcal{M})V, \tag{10}$$

where $\alpha_k$ is the attention weight corresponding to $k$-th bin with $k \in \{1, ..., 4\}$, $A \in \mathbb{R}^{m \cdot m}$ is the attention scores expanded from $\alpha$, $\mathcal{M} \in \{0, -\inf\}$ indicates whether to mask out the value, and $V \in \mathbb{R}^{m \cdot m \times d}$ is the value features sampled from $r'$. By utilizing the mask prediction from previous decoder layer, masked instance-attention can effectively capture objects of different shapes.

### 3.4 Network architecture

SimPLR follows the common end-to-end detection and segmentation framework with a two-stage design (Carion et al., 2020; Zhu et al., 2021; Nguyen et al., 2022). Specifically, we use a plain ViT as the backbone with $14 \times 14$ windowed attention and four equally-distributed global attention blocks as in Li et al. (2022). In the detection head, the SimPLR encoder receives input features via a projection of the last feature map from the ViT backbone. The object proposals are then generated using single-scale features from the encoder and top-scored features are initialized as object queries for the SimPLR decoder to predict bounding boxes and masks.

Formally, we apply a projection $f$ to the last feature map of the pre-trained ViT backbone, resulting in the input feature map $e \in \mathbb{R}^{H_e \times W_e \times d}$ where $H_e, W_e$ are the size of the feature map, and $d$ is the hidden dimension of the detection head. In SimPLR, the projection $f$ is simply a single convolution projection, that provides us a great flexibility to control the resolution and dimension of the input features $e$ to the encoder. The projection allows SimPLR to decouple the feature scale and dimension between its backbone and detection head to further improve the efficiency. This practice is different from the creation of SimpleFPN in Li et al. (2022) where a different stack of multiple convolution layers is used for each feature scale. We show by experiments that this formulation is key for plain detection and segmentation while keeping our network efficient.

**Plain backbone.** SimPLR deploys ViT as its plain backbone for feature extraction. We show that SimPLR can take advantages of recent progress in self-supervised learning and scaling ViTs. To be specific, SimPLR generalizes to ViT backbones initialized by MAE (He et al., 2022) and BEiTv2 (Peng et al., 2022). The efficient design of SimPLR allows us to effectively scale to larger ViT backbones which recently show to be even more powerful in learning representations (He et al., 2022; Zhai et al., 2022; Dehghani et al., 2023).

## 4 Implementation Details

**The creation of input features.** In Fig. 4, we compare the creation of input features to detection head between SimpleFPN and our method. In Li et al. (2022), the multi-scale feature maps are created by different sets of convolution layers. Instead, SimPLR simply applies a deconvolution layer following by a GroupNorm layer (Wu & He, 2018).

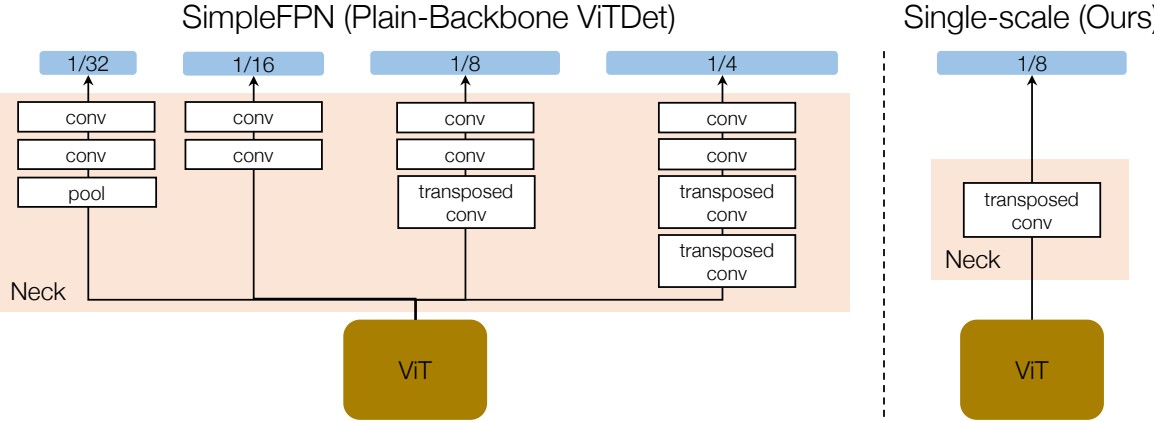

Figure 4: **The creation of input features. Left:** The creation of feature pyramids from the last feature of the plain backbone, ViT, in SimpleFPN (Li et al., 2022) where different stacks of convolutional layers are used to create features at different scales. **Right:** The design of our single-scale feature map with only one layer.

**Losses in training of SimPLR.** We use focal loss (Lin et al., 2017) and dice loss (Milletari et al., 2016) for the mask loss: $\mathcal{L}_{\text{mask}} = \lambda_{\text{focal}}\mathcal{L}_{\text{focal}} + \lambda_{\text{dice}}\mathcal{L}_{\text{dice}}$ with $\lambda_{\text{focal}} = \lambda_{\text{dice}} = 5.0$. The box loss is the combination of $\ell_1$ loss and GIoU loss (Rezatofighi et al., 2019), $\mathcal{L}_{\text{box}} = \lambda_{\ell_1}\mathcal{L}_{\ell_1} + \lambda_{\text{giou}}\mathcal{L}_{\text{giou}}$, with $\lambda_{\ell_1} = 5.0$ and $\lambda_{\text{giou}} = 2.0$. The focal loss is also used for our classification loss, $\mathcal{L}_{\text{cls}}$. Our final loss is formulated as: $\mathcal{L} = \mathcal{L}_{\text{mask}} + \mathcal{L}_{\text{box}} + \lambda_{\text{cls}}\mathcal{L}_{\text{cls}}$ ($\lambda_{\text{cls}} = 2.0$ for object detection and instance segmentation, $\lambda_{\text{cls}} = 4.0$ for panoptic segmentation).

**Hyper-parameters of SimPLR.** SimPLR contains 6 encoder and decoder layers. The adaptive-scale attention in SimPLR encoder samples $2 \times 2$ grid features per region of interest. In the decoder, we compute masked instance-attention on a grid of $14 \times 14$ features within regions of interest. The dimension ratio of feed-forward sub-layers is 4. The number of object queries is 300 in the decoder as suggested in Nguyen et al. (2022). The size of the input image is $1024 \times 1024$ in both training and inference. Note that we also use this setting for the baseline (*i.e.*, BoxeR with ViT backbone).

| model size | backbone | | | detection head | | | |
|---|---|---|---|---|---|---|---|
| | dim. | # heads | feature scale | encoder dim. | decoder dim. | # heads | feature scale |
| Base | 768 | 12 | $\frac{1}{16}$ | 384 | 256 | 12 | $\frac{1}{8}$ |
| Large | 1024 | 16 | $\frac{1}{16}$ | 768 | 384 | 16 | $\frac{1}{8}$ |
| Huge | 1280 | 16 | $\frac{1}{16}$ | 960 | 384 | 16 | $\frac{1}{8}$ |

Table 1: Hyper-parameters of backbone and detection head for different sizes of SimPLR (base – large – huge models). Note that these settings are the same for all three tasks.

| Method | FPS | $\Delta$ FPS [%] | $AP^b$ | $AP^m$ |
|---|---|---|---|---|
| **Feature pyramids** | | | | |
| DETR (Carion et al., 2020) | 15 | 125% | 46.5 | n/a |
| DeformableDETR (Zhu et al., 2021) | 12 | 100% | 54.6 | n/a |
| BoxeR (Nguyen et al., 2022) | 12 | 100% | 55.4 | 47.7 |
| **Plain detector** | | | | |
| PlainDETR (Lin et al., 2023) | 12 | 100% | 53.8 | n/a |
| SimPLR | **17** | **140%** | **55.7** | **47.9** |

Table 2: **SimPLR is an effective end-to-end detector.** Our comparison is between SimPLR with adaptive-scale attention and other end-to-end detectors (*e.g.*, BoxeR with box-attention, DeformableDETR with deformable attention, and PlainDETR with self-attention). For a fair comparison, we report results of all methods using the same ViT-B pre-trained with MAE as backbone. For methods that take feature pyramids as input, we employ SimpleFPN with ViT from (Li et al., 2022). Despite being a plain detector, SimPLR shows competitive performance compared to multi-scale alternatives, while being 40% faster during inference.

In the experiment, we show that the *decouple* between feature scale and dimension of the ViT backbone and the detection head helps to boost the performance of our plain detector while maintaining the efficiency. This comes from the fact that the complexity of global self-attention in the ViT backbone increases quadraticaly w.r.t. the feature scale and the detection head enjoys the high-resolution input for object prediction. Note that with ViT-H as the backbone, we follow Li et al. (2022) to interpolate the kernel of patch projection from the pre-trained $14 \times 14$ into $16 \times 16$. The hyper-parameters for each SimPLR size (Base, Large, and Huge) are in Tab. 1.

## 5 Experiments

**Experimental setup.** We evaluate our method on COCO (Lin et al., 2014), a commonly used dataset for object detection, instance segmentation, and panoptic segmentation tasks. In addition, we also report the performance of our method on panoptic segmentaion using the CityScapes dataset (Cordts et al., 2016). By default, we use plain features with adaptive-scale attention as described in Sec. 3 due to its strong performance and ability to perform both detection and segmentation; and initialize the ViT backbone from MAE (He et al., 2022) pre-trained on ImageNet-1K without any labels. In both fixed-scale and adaptive-scale attention, we set the number of scale $m=4$ and the anchor size $s=32$. Unless specified, our hyper-parameters are the same as in (Nguyen et al., 2022). The frames per second (FPS) of all methods are measured on an A100 GPU. For all experiments, our optimizer is AdamW (Loshchilov & Hutter, 2019) with a learning rate of 0.0001. The learning rate is linearly warmed up for the first 250 iterations and decayed at 0.9 and 0.95 fractions of the total number of training steps by a factor 10. ViT-B (Dosovitskiy et al., 2021) is set as the backbone. The input image size is $1024 \times 1024$ with large-scale jitter (Ghiasi et al., 2021) between a scale range of $[0.1, 2.0]$. Due to the limit of our academic computational resources, we report the ablation study using the standard $5\times$ schedule setting with a batch size of 16 as in Nguyen et al. (2022). In the main experiments, we follow the finetuning recipe from Li et al. (2022).

**SimPLR is an effective end-to-end detector.** In Tab. 2, we first show the comparison between SimPLR and other end-to-end object detectors, with all of them using the plain ViT backbone. Note that SimPLR with scale-aware attention removes the need for multi-scale adaptation of the ViT.

We observe that SimPLR is better than multi-scale end-to-end detectors in both detection and segmentation. Specifically, SimPLR reaches 55.7 in $AP^b$ and 17 frame-per-second using only single-scale input. In terms of accuracy, we outperform DeformableDETR and are slightly better than BoxeR (55.4). Notably, our plain detector is ~40% faster in runtime. We also compare with PlainDETR (Lin et al., 2023) which is a recent plain object detection method. As discussed in Sec. 2, PlainDETR and our work approach the plain detector in different ways. PlainDETR designs a strong decoder that compensates for single-scale input, while our goal is to learn scale equivariant features in the backbone and the encoder. As shown in Tab. 2, SimPLR improves over PlainDETR by ~ 2 AP point in object detection. Moreover, SimPLR also supports instance segmentation, where PlainDETR does not. Despite the different approaches, both PlainDETR and our work indicate that plain detection holds a great potential.

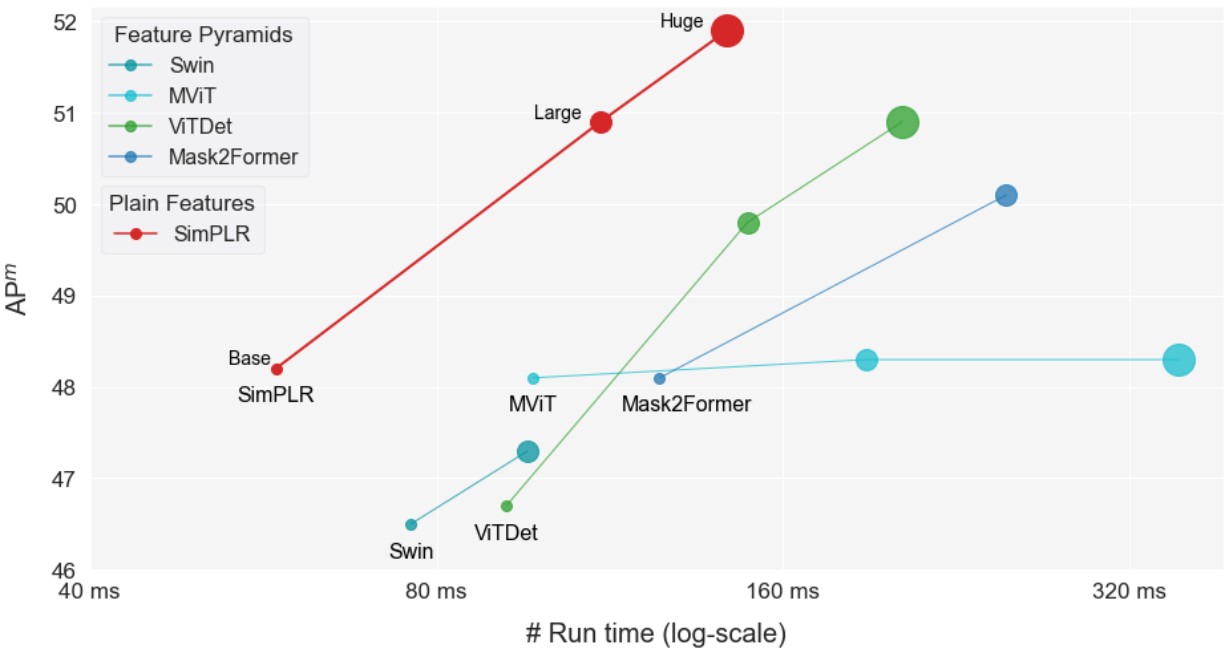

Figure 5: **Scaling comparison.** We compare our plain detector, SimPLR, with recent multi-scale detectors including both end-to-end detector like Mask2Former and plain-backbone detector like ViTDet. Larger circles indicate bigger models (Base, Large, Huge). Because of its plain and simple architecture, SimPLR performance scales with size. Our biggest model achieves stronger performance and faster runtime than all its multi-scale counterparts.

**Scaling comparison.** Fig. 5 illustrates the tradeoffs between accuracy and runtime for various recent detectors, plotted on a log-scale graph. The y-axis represents Average Precision (AP), while the x-axis shows runtime in milliseconds. Different models are indicated using colored markers, with larger circles representing bigger model sizes (Base, Large, Huge). The detectors compared include feature pyramid-based models such as Swin, MViT, ViTDet, and Mask2Former, as well as our plain-feature detector, SimPLR. The graph demonstrates that SimPLR achieves considerable improvements in both accuracy and runtime efficiency as model size increases.

Specifically, the huge model of SimPLR reaches an accuracy of 51.9 AP, maintaining a similar runtime to the large model of ViTDet, which achieves 49.9 AP. Moreover, SimPLR's huge model is faster than Mask2Former, which achieves an accuracy of 50.1 AP in instance segmentation. These results highlight SimPLR's scaling efficiency and effectiveness as a detector. The graph and corresponding data support the conclusion that SimPLR is a scale-efficient detector, offering enhanced performance and faster runtimes compared to its multi-scale counterparts.

**State-of-the-art comparison.** We show in Tab. 3 that SimPLR obtains strong performance on object detection and instance segmentation. When moving to a huge model, our method outperforms multi-scale counterparts like the Mask2Former segmentation model (Cheng et al., 2022) by 1.8 AP point. Despite

| method | backbone | pre-train | Object Detection | | | | Instance Segmentation | | | | FPS |
|---|---|---|---|---|---|---|---|---|---|---|---|
| | | | $AP^b$ | $AP^b_S$ | $AP^b_M$ | $AP^b_L$ | $AP^m$ | $AP^m_S$ | $AP^m_M$ | $AP^m_L$ | |
| **Feature pyramids** | | | | | | | | | | | |
| Swin (Liu et al., 2021) | Swin-L | sup-21K | 55.0 | 38.3 | 59.4 | 71.6 | 47.2 | 28.7 | 50.5 | 66.0 | 10 |
| Mask2Former (Cheng et al., 2022) | Swin-L | sup-21K | | n/a | | | 50.1 | 29.9 | 53.9 | **72.1** | 4 |
| MViT (Fan et al., 2021) | MViT-L | sup-21K | 55.7 | 40.3 | 59.6 | 71.4 | 48.3 | 31.1 | 51.2 | 66.3 | 6 |
| ViTDet (Li et al., 2022) | ViT-L | MAE | 57.6 | 40.5 | 61.6 | 72.6 | 49.9 | 30.5 | 53.3 | 68.0 | 7 |
| MViT (Li et al., 2022) | MViT-H | sup-21K | 55.9 | 40.8 | 59.8 | 70.8 | 48.3 | 30.1 | 51.1 | 66.6 | 6 |
| ViTDet (Li et al., 2022) | ViT-H | MAE | 58.7 | 41.9 | 63.0 | 73.9 | 50.9 | 32.0 | 54.3 | 68.9 | 5 |
| **Plain features** | | | | | | | | | | | |
| SimPLR | ViT-L | MAE | 58.5 | 42.2 | 62.5 | 73.4 | 50.6 | 32.1 | 54.2 | 69.8 | 9 |
| SimPLR | ViT-L | BEiTv2 | 58.7 | 40.4 | 63.2 | 74.8 | 50.9 | 30.4 | 55.1 | 70.9 | 9 |
| SimPLR | ViT-H | MAE | **59.8** | **42.2** | **63.8** | **74.9** | **51.9** | **32.2** | **55.7** | 71.0 | 7 |

Table 3: **State-of-the-art comparison and scaling behavior for object detection and instance segmentation.** We compare methods using feature pyramids *vs.* plain features on COCO `val` (n/a: entry is not available). Backbones with MAE pre-trained on ImageNet-1K while others pre-trained on ImageNet-21K. With only single-scale features, SimPLR shows strong performance compared to multi-scale detectors including transformer-based detectors like Mask2Former and the plain-backbone detector like ViTDet.

| method | backbone | pre-train | COCO | | | CityScapes | FPS |
|---|---|---|---|---|---|---|---|
| | | | PQ | $PQ^{th}$ | $PQ^{st}$ | PQ | |
| **Feature pyramids** | | | | | | | |
| Panoptic FCN (Li et al., 2021) | Swin-L | sup-21K | - | - | - | 65.9 | - |
| Panoptic DeepLab (Cheng et al., 2020) | SWideRNet | sup-1K | - | - | - | 66.4 | - |
| MaskFormer (Cheng et al., 2021) | Swin-B | sup-1K | 51.1 | 56.3 | 43.2 | - | - |
| MaskFormer (Cheng et al., 2021) | Swin-B | sup-21K | 51.8 | 56.9 | 44.1 | - | - |
| Mask2Former (Cheng et al., 2022) | Swin-B | sup-1K | 55.1 | 61.0 | 46.1 | - | 7 |
| Mask2Former (Cheng et al., 2022) | Swin-B | sup-21K | 56.4 | 62.4 | 47.3 | 66.1 | 7 |
| Mask2Former (Cheng et al., 2022) | Swin-L | sup-21K | 57.8 | 64.2 | 48.1 | 66.6 | 4 |
| **Plain features** | | | | | | | |
| SimPLR | ViT-B | sup-1K | 55.5 | 61.4 | 46.2 | - | **13** |
| SimPLR | ViT-B | BEiTv2 | 56.7 | 62.8 | 47.4 | 66.7 | **13** |
| SimPLR | ViT-L | BEiTv2 | **58.8** | **65.3** | **48.8** | **67.4** | 8 |

Table 4: **State-of-the-art comparison and scaling behavior for panoptic segmentation.** We compare between methods using feature pyramids (*top* row) *vs.* single-scale (*bottom* row) on COCO `val` and CityScapes `val`. FPS is reported on COCO. SimPLR with single-scale input shows better results when scaling to larger backbones, while being $\sim 2\times$ faster compared to Mask2Former.

involving more advanced attention blocks designs, *i.e.*, shifted window attention in Swin (Liu et al., 2021) and pooling attention in MViT (Fan et al., 2021), detectors with hierarchical backbones benefit less from larger backbones. SimPLR is better than the plain-backbone detector, ViTDet, across all backbones in terms of both accuracy (by $\sim 1$ AP point) and inference speed. It also worth noting that SimPLR provides good predictions on *small* objects for both detection and segmentation, despite its reliance on single-scale input only.

Tab. 4 shows the comparison for the panoptic segmentation on COCO and CityScapes datasets. We observe the similar trend that SimPLR indicates good scaling behavior. Specifically, in the large model, SimPLR outperforms Mask2Former by 1 PQ point while running $2\times$ faster. We also explore the influence of supervised pre-training on the plain ViT backbone. To keep it simple, we compared Mask2Former with Swin-B and SimPLR with ViT-B both using supervised pre-training on ImageNet-1K. We find that SimPLR still indicates stronger results when using only single-scale input.

**Ablation on SimPLR pre-training data and strategies.** Tab. 5 compares the ViT backbone of SimPLR when pre-trained using different strategies with varying sizes of pre-training data. As expected, SimPLR with

| data | strategy | Object Detection | | | | Instance Segmentation | | | |
|---|---|---|---|---|---|---|---|---|---|
| | | $\text{AP}^b$ | $\text{AP}^b_S$ | $\text{AP}^b_M$ | $\text{AP}^b_L$ | $\text{AP}^m$ | $\text{AP}^m_S$ | $\text{AP}^m_M$ | $\text{AP}^m_L$ |
| **Supervised pre-training** | | | | | | | | | |
| IN-1K | DEiT | 53.6 | 33.7 | 58.1 | 71.5 | 46.1 | 24.5 | 50.4 | 67.2 |
| IN-1K | DEiTv3 | 54.0 | 34.3 | 58.8 | 70.5 | 46.4 | 24.8 | 51.1 | 66.7 |
| IN-21K | DEiTv3 | 54.8 | 35.4 | 59.0 | **72.4** | 47.1 | 25.8 | 51.2 | 68.5 |
| **Self-supervised pre-training** | | | | | | | | | |
| IN-1K | MAE | 55.4 | 36.1 | 59.1 | 70.9 | 47.6 | **26.8** | 51.4 | 67.1 |
| IN-21K | BEiTv2 | **55.7** | **36.5** | **60.2** | **72.4** | **48.1** | 26.7 | **52.7** | **68.9** |

Table 5: **Ablation on SimPLR pre-training data and strategies** with the plain ViT-B backbone evaluated on COCO object detection and instance segmentation. We compare the plain backbone pre-trained using supervised methods (*top* row) *vs.* self-supervised methods (*bottom* row) with different sizes of pre-training dataset (ImageNet-1K *vs.* ImageNet-21K). It can be seen that SimPLR with the plain ViT backbone benefits more pre-training data (*e.g.*, ImageNet-1K *vs.* ImageNet-21K) and better pre-training approaches (*e.g.*, supervised learning *vs.* self-supervised learning). Here, we use the 5× schedule.

| attention | $\text{AP}^b$ | $\text{AP}^m$ |
|---|---|---|
| base | 53.6 | 46.1 |
| fixed-scale | 55.0 | 47.2 |
| adaptive-scale | 55.4 | 47.6 |
| | | |

(a) **Scale-aware attention.**

| $s$ | $\text{AP}^b$ | $\text{AP}^m$ |
|---|---|---|
| base | 53.6 | 46.1 |
| 16 | 55.1 | 47.4 |
| 32 | 55.4 | 47.6 |
| 64 | 55.1 | 47.4 |

(b) **Anchor size.**

| $n$ | $\text{AP}^b$ | $\text{AP}^m$ |
|---|---|---|
| base | 53.6 | 46.1 |
| 2 | 54.6 | 47.0 |
| 4 | 55.4 | 47.6 |
| 6 | 55.2 | 47.6 |

(c) **Number of anchor scales**.

| scale | $\text{AP}^b$ | $\text{AP}^m$ |
|---|---|---|
| base | 53.6 | 46.1 |
| 1/4 | 55.4 | 47.7 |
| 1/8 | 55.4 | 47.6 |
| 1/16 | 54.3 | 46.7 |

(d) **Scales of input features.**

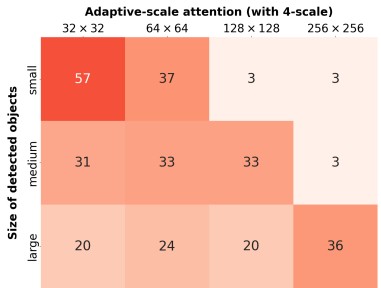

(e) Visualization of scale distribution learnt in **multi-head adaptive-scale attention** of object proposals. Objects are classified into *small*, *medium*, and *large* based on their area.

Table 6: **Ablation of scale-aware attention** in SimPLR using a plain ViT backbone on COCO `val`. **Tables (a-d):** Compared to the baseline, which employs BoxeR and box-attention (Nguyen et al., 2022) with *single-scale* features, SimPLR, with scale-aware attention improves performance consistently for all settings (default setting highlighted). **Figure (e):** our adaptive-scale attention captures different scale distribution in its attention heads based on the context of query vectors. Specifically, queries of *small* objects tends to focus on anchors of small scales (*i.e.*, mainly $32 \times 32$), while query vectors of *medium* and *large* objects distribute more attention computation into larger anchors. All experiments are with 5× schedule.

the ViT backbone benefits from better supervised pre-training methods. Among them, DEiTv3 (Touvron et al., 2022) shows better results than DEiT (Touvron et al., 2021), and pre-training on more data (*i.e.*, ImageNet-21K) further improves DEiTv3 performance. SimPLR with the ViT backbone benefits even more from self-supervised methods like MAE (He et al., 2022), which already provide strong pre-trained backbones when only pre-trained on ImageNet-1K. The best performance is reported with self-supervised method, BEiTv2 (Peng et al., 2022) with more pre-training data of ImageNet-21K. This further confirms that SimPLR, profits from the progress of self-supervised learning and scaling ViTs with more data.

**Ablation of scale-aware attention.** Next, we ablate the scale-aware attention mechanism. As baseline we compare to the standard box-attention from Nguyen et al. (2022) with single-scale feature input directly taken from the last feature of the ViT backbone (denoted as "base").

From Tab. 6a, we first conclude that *both* scale-aware attention strategies are substantially better than the baseline, increasing AP by up to 1.8 points. We note that while fixed-scale attention distributes 25% of its attention heads into each of the anchor scales, adaptive-scale attention decides the scale distribution based on the query content. By choosing feature grids from different scales adaptively, the adaptive-scale attention is able to learn a suitable scale distribution through training data, yielding better performance compared to fixed-scale attention. This is also verified in Fig. 6e where queries corresponding to *small* objects tend to pick anchors of small sizes for its attention heads. Interestingly, queries corresponding to *medium* and *large* objects pick not only anchors of their sizes, but also ones of smaller sizes. A reason may be that performing instance segmentation of larger objects still requires the network to faithfully preserve the per-pixel spatial details.

In Tab. 6b, we compare the performance of SimPLR across several sizes ($s$) of the anchor. They all improve over the baseline, while the choice of a specific base size makes only marginal differences. Our ablation reveals that the number of scales rather than the anchor size plays an important role to make our network more *scale-aware*. Indeed, in Tab. 6c, the use of 4 or more anchor scales shows improvement up to 0.8 AP over 2 anchor scales; and clearly outperforms the naïve baseline. Last, we show in Tab. 6d that the *decouple* between feature scale and dimension of the ViT backbone and the detection head features helps to boost the performance of our plain detector by ∼1 AP point, while keeping its efficiency. This practice makes scaling of SimPLR to larger ViT backbones more practical.

We provide qualitative results for object detection, instance segmentation, and panoptic segmentation on the COCO dataset in Fig. 6 and panoptic segmentaion on the CityScapes dataset in Fig. 7.

**Limitations.** Our final goal is to simplify the detection pipeline and to achieve competitive results at the same time. We find that the *adaptive-scale* attention mechanism that adaptively learns scale-aware information in its computation plays a key role for a plain detector. However, our adaptive-scale attention still encodes the knowledge of different scales. In the future, we hope that with the large-scale training data, a simpler design of the attention mechanism could also learn the scale equivariant property. Furthermore, SimPLR faces difficulties in detecting and segmenting large objects in the image. To overcome this limitation, we think that a design of attention computation which effectively combines both global and local information is necessary.

## 6 Conclusion

We presented SimPLR, a simple and plain object detector that eliminates the requirement for handcrafting multi-scale feature maps. Through our experiments, we demonstrated that a transformer-based detector, equipped with a scale-aware attention mechanism, can effectively learn scale-equivariant features through data. The efficient design of SimPLR allows it to take advantages of significant progress in scaling ViTs, reaching highly competitive performance on three tasks on COCO: object detection, instance segmentation, and panoptic segmentation. This finding suggests that many handcrafted designs for convolution neural network in computer vision could be removed when moving to transformer-based architecture. We hope this study could encourage future exploration in simplifying neural network architectures especially for dense vision tasks.

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

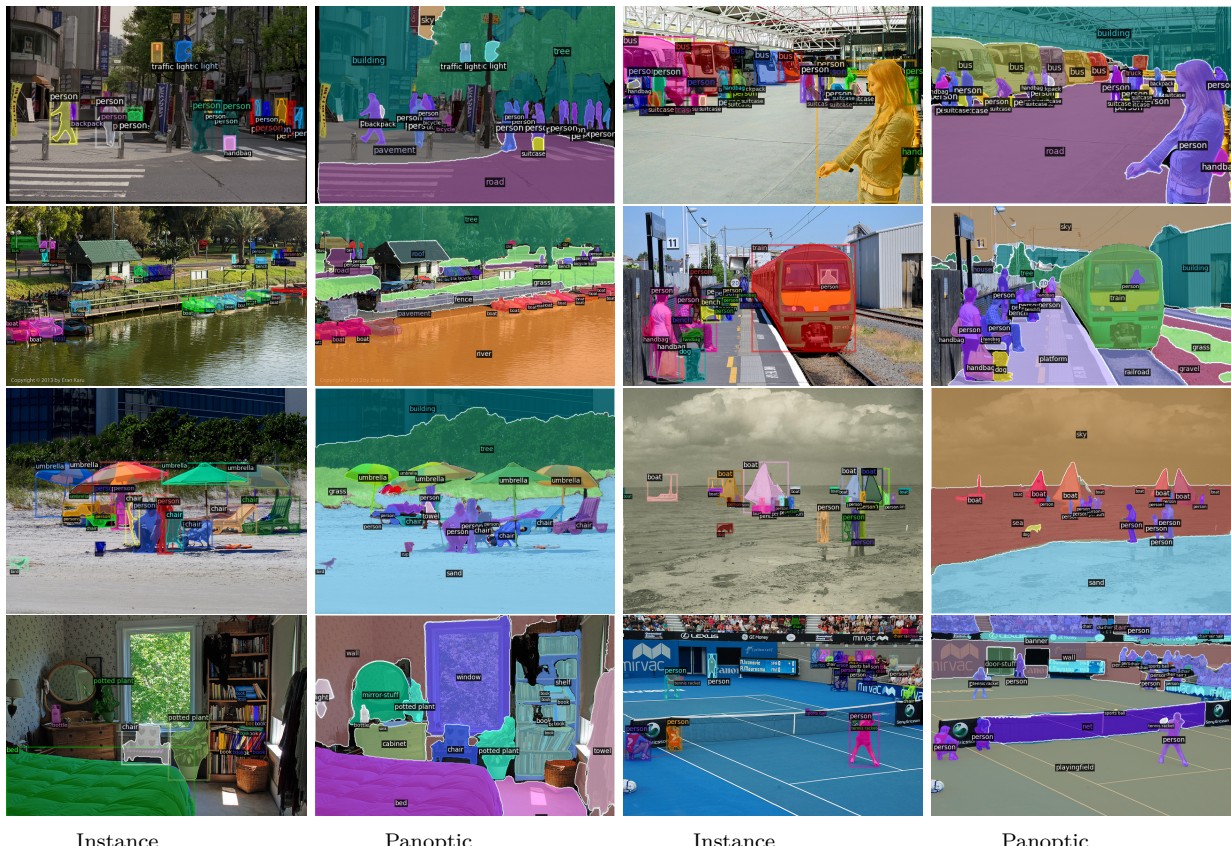

| Instance | Panoptic | Instance | Panoptic |

Figure 6: **Qualitative results** for SimPLR instance detection and segmentation (left) and panoptic segmentation (right) on the COCO `val` set. SimPLR gives good predictions on *small* objects (*e.g.*, the first three rows). However, it is still challenging for the model to detect individual objects in the crowded scenes (*e.g.*, the last row).

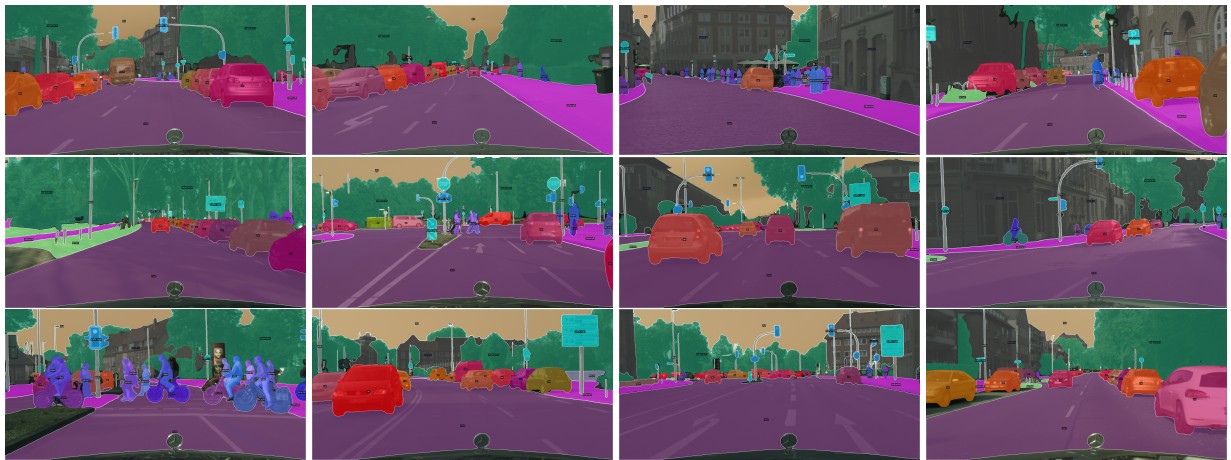

Figure 7: **Qualitative results** of panoptic segmentation generated by SimPLR on Cityscapes `val`. It can be seen that SimPLR correctly predicts *small* objects like pedestrians in various scenes.

Christopher Berner, Sam McCandlish, Alec Radford, Ilya Sutskever, and Dario Amodei. Language models are few-shot learners. In *NeurIPS*, 2020.

Nicolas Carion, Francisco Massa, Gabriel Synnaeve, Nicolas Usunier, Alexander Kirillov, and Sergey Zagoruyko. End-to-end object detection with transformers. In *ECCV*, 2020.

Mark Chen, Alec Radford, Rewon Child, Jeff Wu, Heewoo Jun, Prafulla Dhariwal, David Luan, and Ilya Sutskever. Generative pretraining from pixels. In *ICML*, 2020a.

Ting Chen, Simon Kornblith, Mohammad Norouzi, and Geoffrey Hinton. A simple framework for contrastive learning of visual representations. In *ICML*, 2020b.

Wuyang Chen, Xianzhi Du, Fan Yang, Lucas Beyer, Xiaohua Zhai, Tsung-Yi Lin, Huizhong Chen, Jing Li, Xiaodan Song, Zhangyang Wang, and Denny Zhou. A simple single-scale vision transformer for object localization and instance segmentation. In *ECCV*, 2022.

Bowen Cheng, Maxwell D Collins, Yukun Zhu, Ting Liu, Thomas S Huang, Hartwig Adam, and Liang-Chieh Chen. Panoptic-deeplab: A simple, strong, and fast baseline for bottom-up panoptic segmentation. In *CVPR*, 2020.

Bowen Cheng, Alexander G. Schwing, and Alexander Kirillov. Per-pixel classification is not all you need for semantic segmentation. In *NeurIPS*, 2021.

Bowen Cheng, Ishan Misra, Alexander G. Schwing, Alexander Kirillov, and Rohit Girdhar. Mask2former: Masked-attention mask transformer for universal image segmentation. In *CVPR*, 2022.

M. Cordts, M. Omran, S. Ramos, T. Rehfeld, M. Enzweiler, R. Benenson, U. Franke, S. Roth, and B. Schiele. The cityscapes dataset for semantic urban scene understanding. In *CVPR*, 2016.

Mostafa Dehghani, Josip Djolonga, Basil Mustafa, and et al. Scaling vision transformers to 22 billion parameters. In *ICML*, 2023.

Jia Deng, Wei Dong, Richard Socher, Li-Jia Li, Kai Li, and Li Fei-Fei. Imagenet: A large-scale hierarchical image database. In *CVPR*, 2009.

Jacob Devlin, Ming-Wei Chang, Kenton Lee, and Kristina Toutanova. Bert: Pre-training of deep bidirectional transformers for language understanding. In *ACL*, 2019.

Bin Dong, Fangao Zeng, Tiancai Wang, Xiangyu Zhang, and Yichen Wei. SOLQ: Segmenting objects by learning queries. In *NeurIPS*, 2021.

Alexey Dosovitskiy, Lucas Beyer, Alexander Kolesnikov, Dirk Weissenborn, Xiaohua Zhai, Thomas Unterthiner, Mostafa Dehghani, Matthias Minderer, Georg Heigold, Sylvain Gelly, Jakob Uszkoreit, and Neil Houlsby. An image is worth 16x16 words: Transformers for image recognition at scale. In *ICLR*, 2021.

Haoqi Fan, Bo Xiong, Karttikeya Mangalam, Yanghao Li, Zhicheng Yan, Jitendra Malik, and Christoph Feichtenhofer. Multiscale vision transformers. In *ICCV*, 2021.

Golnaz Ghiasi, Yin Cui, Aravind Srinivas, Rui Qian, TsungYi Lin, Ekin D Cubuk, Quoc V Le, and Barret Zoph. Simple copy-paste is a strong data augmentation method for instance segmentation. In *CVPR*, 2021.

Ross Girshick, Jeff Donahue, Trevor Darrell, and Jitendra Malik. Rich feature hierarchies for accurate object detection and semantic segmentation. In *CVPR*, 2014.

Kaiming He, Xiangyu Zhang, Shaoqing Ren, and Jian Sun. Deep residual learning for image recognition. In *CVPR*, 2016.

Kaiming He, Georgia Gkioxari, Piotr Dollár, and Ross Girshick. Mask R-CNN. In *ICCV*, 2017.

Kaiming He, Xinlei Chen, Saining Xie, Yanghao Li, Piotr Dollár, and Ross Girshick. Masked autoencoders are scalable vision learners. In *CVPR*, 2022.

Byeongho Heo, Sangdoo Yun, Dongyoon Han, Sanghyuk Chun, Junsuk Choe, and Seong Joon Oh. Rethinking spatial dimensions of vision transformers. In *ICCV*, 2021.

Zhengdong Hu, Yifan Sun, Jingdong Wang, and Yi Yang. DAC-DETR: Divide the attention layers and conquer. In *NeurIPS*, 2023.

Gao Huang, Zhuang Liu, Laurens van der Maaten, and Kilian Q. Weinberger. Densely connected convolutional networks. In *CVPR*, 2017.

Ding Jia, Yuhui Yuan, Haodi He, Xiaopei Wu, Haojun Yu, Weihong Lin, Lei Sun, Chao Zhang, and Han Hu. Detrs with hybrid matching. In *CVPR*, 2023.

Alexander Kirillov, Kaiming He, Ross Girshick, Carsten Rother, and Piotr Dollar. Panoptic segmentation. In *CVPR*, 2019.

Harold W. Kuhn. The Hungarian method for the assignment problem. *Naval Res. Logist. Quart*, 1955.

Yann LeCun and Yoshua Bengio. *Convolutional Networks for Images, Speech and Time Series*, pp. 255–258. MIT Press, 1995.

Feng Li, Hao Zhang, Huaizhe xu, Shilong Liu, Lei Zhang, Lionel M. Ni, and Heung-Yeung Shum. Mask dino: Towards a unified transformer-based framework for object detection and segmentation. In *CVPR*, 2023.

Yanghao Li, Hanzi Mao, Ross Girshick, and Kaiming He. Exploring plain vision transformer backbones for object detection. In *ECCV*, 2022.

Yanwei Li, Hengshuang Zhao, Xiaojuan Qi, Liwei Wang, Zeming Li, Jian Sun, , and Jiaya Jia. Fully convolutional networks for panoptic segmentation. In *CVPR*, 2021.

Tsung-Yi Lin, Michael Maire, Serge J. Belongie, Lubomir D. Bourdev, Ross B. Girshick, James Hays, Pietro Perona, Deva Ramanan, Piotr Dollár, and C. Lawrence Zitnick. Microsoft COCO: common objects in context. In *ECCV*, 2014.

Tsung-Yi Lin, Piotr Dollár, Ross Girshick, Kaiming He, Bharath Hariharan, and Serge Belongie. Feature pyramid networks for object detection. In *CVPR*, 2017.

Tsung-Yi Lin, Priya Goyal, Ross B. Girshick, Kaiming He, and Piotr Dollár. Focal loss for dense object detection. In *ICCV*, 2017.

Yutong Lin, Yuhui Yuan, Zheng Zhang, Chen Li, Nanning Zheng, and Han Hu. DETR does not need multi-scale or locality design. In *ICCV*, 2023.

Wei Liu, Dragomir Anguelov, Dumitru Erhan, Christian Szegedy, Scott Reed, Cheng-Yang Fu, and Alexander C. Berg. SSD: Single shot multibox detector. In *ECCV*, 2016.

Ze Liu, Yutong Lin, Yue Cao, Han Hu, Yixuan Wei, Zheng Zhang, Stephen Lin, and Baining Guo. Swin transformer: Hierarchical vision transformer using shifted windows. In *ICCV*, 2021.

Ilya Loshchilov and Frank Hutter. Decoupled weight decay regularization. In *ICLR*, 2019.

Fausto Milletari, Nassir Navab, and Seyed-Ahmad Ahmadi. V-net: Fully convolutional neural networks for volumetric medical image segmentation. In *3DV*, 2016.

Duy-Kien Nguyen, Jihong Ju, Olaf Booij, Martin R. Oswald, and Cees G. M. Snoek. Boxer: Box-attention for 2d and 3d transformers. In *CVPR*, 2022.

Zhiliang Peng, Li Dong, Hangbo Bao, Qixiang Ye, and Furu Wei. BEiT v2: Masked image modeling with vector-quantized visual tokenizers. *arXiv preprint arXiv:2208.06366*, 2022.

Shaoqing Ren, Kaiming He, Ross Girshick, and Jian Sun. Faster R-CNN: Towards real-time object detection with region proposal networks. In *NeurIPS*, 2015.

Seyed Hamid Rezatofighi, Nathan Tsoi, JunYoung Gwak, Amir Sadeghian, Ian D. Reid, and Silvio Savarese. Generalized intersection over union: A metric and A loss for bounding box regression. In *CVPR*, 2019.

Karen Simonyan and Andrew Zisserman. Very deep convolutional networks for large-scale image recognition. In *ICLR*, 2015.

Hugo Touvron, Matthieu Cord, Matthijs Douze, Francisco Massa, Alexandre Sablayrolles, and Herve Jegou. Training data-efficient image transformers and distillation through attention. In *International Conference on Machine Learning*, 2021.

Hugo Touvron, Matthieu Cord, and Herve Jegou. Deit iii: Revenge of the vit. *arXiv preprint arXiv:2204.07118*, 2022.

Ashish Vaswani, Noam Shazeer, Niki Parmar, Jakob Uszkoreit, Llion Jones, Aidan N Gomez, Lukasz Kaiser, and Illia Polosukhin. Attention is all you need. In *NeurIPS*, 2017.

Huiyu Wang, Yukun Zhu, Hartwig Adam, Alan Yuille, and Liang-Chieh Chen. Max-deeplab: End-to-end panoptic segmentation with mask transformers. In *CVPR*, 2021a.

Wenhai Wang, Enze Xie, Xiang Li, Deng-Ping Fan, Kaitao Song, Ding Liang, Tong Lu, Ping Luo, and Ling Shao. Pyramid vision transformer: A versatile backbone for dense prediction without convolutions. In *ICCV*, 2021b.

Yuxin Wu and Kaiming He. Group normalization. In *ECCV*, 2018.

Saining Xie, Ross B. Girshick, Piotr Dollár, Zhuowen Tu, and Kaiming He. Aggregated residual transformations for deep neural networks. In *CVPR*, 2017.

Qihang Yu, Huiyu Wang, Siyuan Qiao, Maxwell Collins, Yukun Zhu, Hartwig Adam, Alan Yuille, and Liang-Chieh Chen. k-means mask transformer. In *ECCV*, 2022.

Xiaohua Zhai, Alexander Kolesnikov, Neil Houlsby, and Lucas Beyer. Scaling vision transformers. In *CVPR*, 2022.

Hao Zhang, Feng Li, Shilong Liu, Lei Zhang, Hang Su, Jun Zhu, Lionel Ni, and Heung-Yeung Shum. Dino: Detr with improved denoising anchor boxes for end-to-end object detection. In *ICLR*, 2023.

Wenwei Zhang, Jiangmiao Pang, Kai Chen, and Chen Change Loy. K-net: Towards unified image segmentation. In *NeurIPS*, 2021.

Xizhou Zhu, Weijie Su, Lewei Lu, Bin Li, Xiaogang Wang, and Jifeng Dai. Deformable DETR: Deformable transformers for end-to-end object detection. In *ICLR*, 2021.

Zhuofan Zong, Guanglu Song, and Yu Liu. Detrs with collaborative hybrid assignments training. In *ICCV*, 2023.

