# OpenReview forum: "SimPLR: A Simple and Plain Transformer for Efficient Object Detection and Segmentation"
_TMLR — Accepted by TMLR_

### Review · Reviewer_ajHh · 2024-08-31

**Summary Of Contributions:**

The main contribution of this paper lies in the introduction of a scale-attention scheme (multi-head scale-aware) aimed at enhancing the multi-scale capability of a given single-scale feature map. According to the description in Section 3.2 and illustrated in Figure 2, the proposed adaptive-scale attention can be seen as a simplification of the more powerful multi-scale deformable attention proposed in the previous work, Deformable DETR [1]. Deformable DETR predicts adaptive offsets not only across different attention heads but also across various feature map scales.
Regarding the masked instance-attention, it is fundamentally the same as the masked attention mechanism introduced in Mask2Former [2]. Additionally, the proposed use of a single-scale feature map based on a plain ViT has been thoroughly investigated in Section 5.4 of the prior work, PlainDETR [3].
The effectiveness of the proposed method is not convincingly demonstrated. Based on my experience, the observed gains over previous methods such as Deformable DETR, Mask2Former, and PlainDETR seem to stem more from the selection of superior pre-trained backbones or the deformable attention design, like BEiTv2 and DeformableDETR, rather than the proposed scale-attention scheme.


[1] Deformable DETR: Deformable Transformers for End-to-End Object Detection, ICLR’2021

[2] Masked-attention Mask Transformer for Universal Image Segmentation, CVPR’2022

[3] DETR Does Not Need Multi-Scale or Locality Design, ICCV’2023

**Audience:**

Yes

**Broader Impact Concerns:**

There are no concerns regarding the ethical implications of this work.

**Claims And Evidence:**

No

**Requested Changes:**

The authors should enhance this work by designing a novel and effective module that genuinely outperforms previous approaches or by offering new insights to the community. The current version fails to meet these criteria. Additionally, the authors should address the aforementioned weaknesses.

**Strengths And Weaknesses:**

> Strengths

- The motivation behind this work is compelling, as it explores a promising direction focused on developing simple and straightforward architectural designs for fundamental visual recognition tasks, including object detection and panoptic segmentation.
- The writing is clear, and the proposed scale-aware attention method is simple and easy to follow.

> Weaknesses

- The key design of this work lacks deep technical contributions, and similar ideas have been thoroughly analyzed in previous works such as Deformable DETR, Mask2Former, and Plain-DETR.

- The effectiveness of the proposed method is not convincingly demonstrated. The authors should elucidate why such a simplification of multi-scale deformable attention can outperform previous, more advanced designs. The following major concerns should be addressed:

1. **Efficiency Comparison (Table 2)**: Why is SimPLR more efficient than Plain-DETR? Are the comparisons fair? How does the choice of backbone and the number of parameters affect the results? It is acknowledged that the plain Transformer decoder design is less efficient than Deformable attention; if this is the primary reason, the advantage is due to the sparsity introduced by Deformable attention.

2. **Performance Comparison (Tables 3 and 4)**: Why does the proposed SimPLR outperform previous methods like ViTDet and Mask2Former? How would the combination of Deformable attention with ViTDet perform? Additionally, the authors should include comparisons with Plain-DETR (AP=60 or 63.9 with or without pre-training on Object 365).

3. **Mathematical Formulation**: The mathematical formulation is confusing. For instance, in the section on adaptive-scale attention, what does $[r_j]_{j=0}^{m-1}$ represent? It is assumed to denote the region proposals associated with different scales. The authors should clarify this notation.

---

> ### Author Response · Authors · 2024-09-16
>
> We thank reviewer for comments appreciating that the proposed scale-aware attention method is ```simple``` and ```easy to follow```, and the writing is ```clear```. We address concerns below and will incorporate discussions in our updated paper.
>
> Overall, we’d like to emphasize the following points:
>
> - Our primary goal is not to focus extensively on architectural design, but rather to enable a sparse attention mechanism for the plain detector.
> - Despite its simplicity, our adaptive-scale attention mechanism for single-scale input delivers competitive performance when compared to multi-scale counterparts, while also being faster and more memory efficient.
> - We evaluate the effectiveness of our plain detector, SimPLR, on three major tasks: object detection, instance segmentation, and panoptic segmentation.
>
> 1. ***Comparisons with Plain-DETR.***
> - While Plain-DETR claims to be a plain detector, it actually relies on a hierarchical backbone (SwinTransformer). When switching to a ViT backbone pre-trained with MAE, its performance drops by more than 1.5 AP points.
> - Furthermore, Plain-DETR continues to use multi-scale features for object proposal prediction, as noted in their official GitHub repository (https://github.com/impiga/Plain-DETR/tree/main?tab=readme-ov-file#limitation–discussion). Generating proposals with plain features leads to an additional 1 AP point performance drop.
> - Plain-DETR is also restricted to object detection, whereas SimPLR excels across three tasks: object detection, instance segmentation, and panoptic segmentation.
> - Additionally, the reported performance of Plain-DETR (AP=60 without pre-training on Object 365) includes the use of test-time augmentation. This contradicts the idea of a plain detector and significantly slows down the detection pipeline.
> - In contrast, we believe that SimPLR is a simple and efficient plain detector that operates purely on plain features while maintaining high performance and being efficient.
>
> 2. ***Why is SimPLR more efficient than Plain-DETR?***:
> - SimPLR is more efficient than Plain-DETR due to the use of spare attention mechanism instead of the self-attention in the plain Transformer design.
> - The key factor behind this efficiency is the sparsity of the attention mechanism. However, it is our goal to develop *a sparse attention mechanism specifically for the plain detector*.
> - As demonstrated in Table 2, our sparse attention mechanism *outperforms* Deformable Attention, even when Deformable Attention is applied to multi-scale features (we use ViT pre-trained with MAE + SimpleFPN in this case). Additionally, when Deformable Attention is applied to plain features, its performance drops even further compared to its multi-scale version.
>
> 3. ***Are the comparisons fair?***
> - Yes, the comparisons are fair. To ensure this, we used the official PlainDETR GitHub repository (https://github.com/impiga/Plain-DETR) and replaced their Swin backbone with the plain ViT backbone combined with SimpleFPN [1].
>
> 4. ***How does the choice of backbone and the number of parameters affect the results?***
> - The impact of backbone choices and the number of parameters is demonstrated in Figure 5 and Tables 3 and 4. As we scale the backbone from ViT-B to ViT-L and ViT-H, SimPLR’s performance improves across all tasks, including object detection, instance segmentation, and panoptic segmentation.
>
> 5. ***Why does the proposed SimPLR outperform previous methods like ViTDet and Mask2Former?***
> - We believe ViTDet’s limitation lies in its convolutional detection head, which performs worse as the model size increases. Additionally, Mask2Former relies on a hierarchical backbone, and its design becomes significantly slower and more memory-inefficient when applied to a plain backbone due to its global attention mechanism.
> - With advancements in pre-training techniques and better scaling behavior, the ViT backbone has become more competitive compared to hierarchical backbones like Swin.
>
> 6. ***How would the combination of Deformable attention with ViTDet perform?***
> - Table 2 (Deformable Attention + SimpleFPN with ViT) reports the performance of combining Deformable Attention with ViTDet, showing that SimPLR clearly outperforms the Deformable Attention + ViTDet combination.
>
> 7. ***Mathematical Formulation***:
> - Apologies for the confusion. We will provide a clearer explanation in the method section.
> - The notation $[r_j]^{m-1}_{j=0}$ represents the set of anchors across $m$ scales, indexed from $0$ to $m-1$. Each anchor in this set is refined into a region proposal by applying the offsets predicted by our scale-aware attention mechanism.

---

> > ### Comment · Reviewer_ajHh · 2024-11-17
> > **Thanks for authors' response**
> >
> > I have carefully reviewed the authors' response and the other reviews. While some of my concerns have been adequately addressed, several fundamental issues remain. For instance:
> >
> > -“According to the description in Section 3.2 and illustrated in Figure 2, the proposed adaptive-scale attention can be viewed as a simplification of the more powerful multi-scale deformable attention presented in previous work, Deformable DETR [1]. Deformable DETR predicts adaptive offsets not only across different attention heads but also across various feature map scales. Additionally, the masked instance-attention is fundamentally similar to the masked attention mechanism introduced in Mask2Former [2].”-
> >
> > The authors have not responded to this question to justify the novel technical contributions of this paper. Therefore, I maintain my original rating that this paper is not yet ready for publication.

---

> ### Author Response · Authors · 2024-11-17
>
> Dear Reviewer,
>
> Thank you for your observation. As previously mentioned, our primary contribution is enabling *a sparse attention mechanism for plain feature maps*.
>
> We respectfully disagree that multi-scale deformable attention is inherently more powerful. Table 2 compares our approach with DeformableDETR using ViT-B with SimpleFPN, demonstrating that our scale-aware attention outperforms deformable attention in both performance and inference speed. Notably, DeformableDETR performs significantly worse on single-scale features (the AP drops by 2 points). This aligns with the observation in the original paper that deformable attention requires multi-scale features.
>
> We believe these improvements of performance and inference speed confirm the value of our work. If you have further questions, we would be happy to address them.
>
> Best regards,
> Authors

---

> > ### Comment · Reviewer_ajHh · 2024-11-19
> > **Clarification on my concern**
> >
> > I agree the authors' claim that the "primary contribution is enabling a sparse attention mechanism for plain feature maps." However, we also need to consider the key novel technical design behind this work.
> >
> > My major concern is that "the proposed adaptive-scale attention can be viewed as a simplification of the more powerful multi-scale deformable attention presented in previous work," and "the masked instance-attention is fundamentally similar to the masked attention mechanism introduced in Mask2Former." Therefore, I expect the authors to clarify whether there is any misunderstanding. In other words, **the key technical contribution of this work appears weak, as it simply combines two techniques introduced in previous works, even though the authors have found that a simplified version works better for plain architectures.**
> >
> > I hope the AE and other reviewers will also consider this aspect for the final rating.

---

> ### Author Response · Authors · 2024-11-19
>
> Dear Reviewer,
>
> Thank you for your thoughtful feedback and for acknowledging our efforts, as noted in your comment: ```Authors have found that a simplified version works better for plain architectures.```
>
> We would like to highlight that the concept of predicting adaptive offsets was initially proposed in [a]. In DeformableDETR, the authors had been successfully adapting it for multi-scale features in end-to-end detection with transformers.
>
> While we agree that our attention mechanism builds on the idea of predicting adaptive offsets, we believe that its ability to effectively learn multi-scale information from plain features is *non-trivial*. This is supported by the extensive experiments presented in our manuscript, which demonstrate significant improvements in both performance and inference speed compared to DeformableDETR.
>
> Best,
>
> Authors
>
> [a] Deformable Convolutional Networks. Dai et al. In ICCV 2017.

---

### Review · Reviewer_TLhE · 2024-10-19

**Summary Of Contributions:**

This paper proposes an approach for object detection and instance/panoptic segmentation following in the general approach of DETR style detectors where a pretrained feature extractor is followed by processing using a transformer encoder/decoder.  Following a series of recent works, this paper replaces the attention in the transformer part of the model by a more targeted form (box attention) where the attention is restricted to grid points sampled from a bounding box.  The novel contribution in this paper is that the bounding boxes are predicted relative to anchors with multiple scales.

This approach multi-scale-in-anchors (but not multi-scale-in-image-features) approach allows them to get strong results without using multiple input resolutions from a feature pyramid and simplifies the architecture somewhat.  And I would note that it is conceptually similar to earlier single stage anchor-based detectors like SSD (Liu et al). Results on standard academic benchmarks are quite strong.  Within DETR style detectors it has competitive (even SOTA) box mAP and mask mAP and is competitive with the SOTA in general and is fast and arguably a simpler approach.

**Audience:**

Yes

**Claims And Evidence:**

Yes

**Requested Changes:**

Overall, addressing the issues raised in the above would strengthen the work in my opinion but are not critical to securing my recommendation for acceptance (I do recommend acceptance).

Some other thoughts/suggestions to consider:
* One question to investigate is whether SimPLR trains faster than prior work since the FPN presumably would require extra time to train after initialization from the base network.
* If I were playing devil’s advocate I would say that there is still more headroom for simplification and the multiscale anchor design is not that simple conceptually — there are details here for example that could easily lead to errors in implementation.  I would also recommend a discussion of what specific further simplifications researchers might want to explore

**Strengths And Weaknesses:**

The main contribution of this work is a simplification of previous multiscale DETR-style pipelines.  The results are strong relative to the state of the art and computationally efficient.  This is a valuable contribution to the community and merits acceptance.

The main weaknesses of this work are writing quality and missing relevant baselines.

On writing:
* The explanation of the proposed method is not well self-contained and requires familiarity with earlier papers (DETR, Deformable DETR, Boxer, etc).  While the first two are fairly well known in the community, I do not think Boxer is as well known and could use more details (i.e. pretend the reader has never read Boxer)
* Fixed scale attention and adaptive scale attention — both of these seem very similar, I’m not sure if I understand why the adaptive approach would intuitively be that much better and they even seem similar in ablations.
* One confusing notational issue is that m here denotes the number of scales while in the original Boxer paper it is used to denotes the resolution of the grid used to sample within a box.  Consider using another letter?
* Masked instance attention is also confusing as the effectiveness of the design is not ablated and it is not motivated.  The 2 × 2 grid seems to come from nowhere... it is never mentioned here and I was only able to find it after looking in the appendix of the Boxer paper.  Perhaps it would be better to write this more generally for kxk grids and offer a motivation for this module?  I would also suggest a more clear picture of this architecture.
* Is there a clearer way to explain what makes the SimPLR approach faster than other approaches?

For relevant baselines (in the state-of-the-art comparisons),
* In what sense is this table a fair comparison across approaches?  Are, e.g. input resolutions fixed across models?
* The CoDETR paper for example reports reaching 65 box AP points...  I don’t think this detracts from whether this paper should be accepted, but should be cited.
* The VitDet paper also achieves better numbers than what is reported in this submission (see table 6 in the VitDet paper)

---

> ### Author Response · Authors · 2024-10-31
>
> Thanks for the careful review and detailed comments, noting our results ```strong```, and our contribution ```valuable```. We address the concerns next and will update our paper accordingly.
>
> 1. ***Fair Comparison Across Approaches***
> - Our detectors are trained on 1024x1024 images without pre-trained data (e.g., Object365), and we avoid advanced techniques like denoising queries (as in DINO) and complex matching methods (e.g., CoDETR, HybridMatching) to reduce approach complexity. These techniques slow training and complicate the use of larger backbones (e.g., ViT-H), making our comparisons to ViTDet and Mask2Former more balanced.
> - For ViTDet, we used official checkpoints from the official github (https://github.com/facebookresearch/detectron2/tree/main/projects/ViTDet) and evaluated them on COCO detection and instance segmentation. These checkpoints were trained at 1024x1024 resolution, as shown in their Table 5 of the ViTDet paper instead of 1280x1280 resolution in Table 6.
>
> 2. ***Speed Advantage of SimPLR***
> - The speed of SimPLR stems from two key factors: sparse attention and single-scale input.
> - Sparse attention allows for linear cost increases with higher resolutions, while single-scale input enhances efficiency when scaling up to larger ViT backbones and higher resolution inputs.
>
> 3. ***Fixed scale attention vs. adaptive scale attention***
> - Both methods aggregate multi-scale information within the attention mechanism. In fixed-scale attention, attention heads are uniformly distributed across scales, while in adaptive scale attention, this distribution is learned from the data, providing greater flexibility.
>
> 4. ***Clarity and Notational Issues***
> - We will revise our draft for clarity.
>
> 5. ***Suggestions to consider***
> - Thanks for your suggestion. As suggested by the reviewer cMeH, our work shows that absorbing multi-scale information into attention mechanism allows us to ease the requirements of multi-scale inputs.
> - We think it may be an exciting direction for future work to explore ways to integrate multi-scale learning directly into standard self-attention (potentially through multi-scale positional information).

---

> ### Author Response · Authors · 2024-12-02
> **Feedbacks from Reviewer TLhE**
>
> Dear reviewer TLhE,
>
> It's been a while since we posted our response. Please let us know if you had a chance to read it and have an updated thought. If anything is less clear, we are happy to address it.

---

### Review · Reviewer_cMeH · 2024-10-20

**Summary Of Contributions:**

This work proposes "a plain detector" for object detection and segmentation, where "plain" means that the backbone is not hierarchichal (e.g. ViTDet) and multiscale feature maps (e.g. Deformable DETR) are not used in the detection/segmentation modules

From the perspective of "engineering contributions":

- Gains in speed and accuracy have been reported, e.g. 1.4x faster than ViT-B based Deformable DETR (+1.1 box AP) and BoxeR (+0.3 box AP) baselines

- SimPLR performs reasonably at larger scale as well, e.g. +1.1 vs VitDet-H on COCO detection and has reasonable performance on panoptic segmentation

**Audience:**

Yes

**Claims And Evidence:**

No

**Requested Changes:**

- Is it possible to add experiments to back some of the scientific claims? e.g. see weakness A. Alternatively, would the authors be ok with relaxing the claims to something more along the lines of "shifting the multiscale inductive bias into the attention mechanism can work well, resulting in speed and accuracy gains"?

 - Would the authors be ok with relaxing the claims of "removing simplifying and removing multiscale inductive biases" (see weakness B)?

- Can FPS numbers wherever relevant, be reported with optimized attention kernels and please add additional details regarding baselines in Figure 5? Can you report the speed of the sprase, scale-aware attention layer vs a regular self-attention layer (using optimized kernels)?

- Can the authors run appriopriate baseline for scaling for Figure 5 to support "more scaling efficient" or alternatively relax the claim to something along the lines of "SimPLR performance scales with size"?

- [Not critical] Can you sweep over hyperparamters for the baselines? (see weakness C)

- Can you report performance on LVIS? (see weakness D)

**Strengths And Weaknesses:**

Strengths:

- A. I believe the "engineering contributions" discussed above are of possible interest to the community and shows that a sparse scale-aware attention can be relatively fast and have good performance vs commonly adopted architecture choices (ViTDet, Deformable DETR)

- B. Presentation is clear

Weaknesses:

 - A. A central claim in the paper is that multiscale feature maps and pyramid designs are _unnecessary_, but there is a lack of experimental evidence for this scientific claim. For e.g. results in Table 2 are all using a ViT-B, a plain, non-hierarhichal backbone and so there isn't a true FPN. If there isn't a comparison to a true hierarchichal backbone (e.g. Swin, Hiera trained w/ MAE) in a controlled experiment setup (as in Table 2), I'm not sure how it can be claimed that multiscale feature maps and pyramid designs are unnecessary - an alternative hypothesis could simple be that SimpleFPN isn't very effective.

Relatedly, another important baseline here is a plain backbone using (single scale) features from multiple layers - after all, an FPN is not just about multiscale features but that high-resolution features come from earlier in the network.

- B. A core motivation of the work is to question the role of and remove hand-crafted inductive biases (such as multiscale feature maps, pyramids) and see how far we can get with the ViT-style approach of learning from data instead of baking such biases in. And yet, as a reader, respectfully, it feels a bit like a bait-and-switch. Instead of having multiscale feature maps, a multiscale bias is instead pushed into the attention mechanism. The distinction that the detector still uses single scale feature maps strikes me as artificial (scientifically speaking).

- C. It worries me that all results for Table 2 use the same hyperparameters and could result in bias. For e.g. ViTDet swept over hyperparams for other methods (e.g. Swin) to mitigate.

- D. I worry how COCO specific (e.g. small number of classes) the detection/instance seg results are. While the alternatives of ViTDet, Swin are known to also work well on LVIS. Having some LVIS number is an important baseline in current literature in detection/instance seg.

- E. Were all the FPS number in Figure 5 run by you? Maybe I missed it what detection framework each of the baselines in the figure are using?

- F. Were all relevant FPS numbers run with optimized attention kernels (e.g. flashattention v2)? e.g. DETR baseline in Table 2. Can the authors compare the speed of their sparse detection head with baselines using flashattention? what's the precision used?

This figure has been used to claim that SimPLR is "more scaling efficient". To support such a claim, I would have expected relevant baselines (e.g. from table 2) scaled as well. As the figure currently stands, I'd read it instead as SimPLR's performance scales with size, rather than SimPLR is more scaling efficient than alternatives.

---

> ### Author Response · Authors · 2024-10-31
>
> Thanks for the detailed and constructive comments to help our work, acknowledging our presentation ```clear```, our approach ```gains in speed``` and ```performs reasonably at larger scale ```. Next, we address the concerns and will include them in our updated paper.
>
> 1. ***The necessity of Multiscale Feature Maps and Pyramid Designs***
> - Thank you for this suggestion. We agree that a more nuanced claim, such as “learning multiscale information within the attention mechanism can be effective,” better represents our findings. We will update the draft accordingly.
>
> 2. ***Hyperparameter in Table 2***
> - Conducting a comprehensive hyperparameter sweep for each baseline is beyond our resource capabilities. To mitigate bias, we did not optimize training hyperparameters specifically for our approach. Instead, we used the standard 5x learning schedule with a batch size of 16, as widely adopted (e.g., Mask2Former, DINO). For architecture-specific hyperparameters, we adhered to each original paper’s optimal settings.
> - Notably, our reported performance for DeformableDETR + ViT with SimpleFPN in Table 2 (54.6 AP) significantly exceeds the 52.1 AP reported in Table 8 of the PlainDETR paper.
>
> 3. ***Detection Frameworks for Baselines in Figure 5***
> - We ran each baseline using its official repository in Detectron2.
>
> 4. ***The use of Optimized Attention Kernels for FPS***
> - We used optimized attention kernels (e.g., F.scaled_dot_product_attention in PyTorch) for the ViT backbone with SimpleFPN across methods. However, we forgot to use optimized kernels in the DETR detection head which results in the FPS of 14 for DETR. Despite this, our approach maintains a faster inference time.
>
> 5. ***Performance on LVIS***
> - We are currently adapting LVIS for our codebase and will report results upon completing our experiments.
> - However, we reported the performance of our approach on CityScapes to further verify its effectiveness.
>
> 6. ***Scaling Baseline for Figure 5"***
> - Since PlainDETR authors only released their model with a small backbone, our attempts to train it on a larger backbone showed slow and unstable training. Thus, we kept our baseline as Mask2Former (representative of hierarchical backbone + multi-scale) and ViTDet (representative of plain backbone + multi-scale).

---

> ### Author Response · Authors · 2024-12-02
> **Feedbacks from Reviewer cMeH**
>
> Dear reviewer cMeH,
>
> It's been a while since we posted our response. Please let us know if you had a chance to read it and have an updated thought. If anything is less clear, we are happy to address it.

---

### Decision · Action_Editor_F2QS · 2025-01-17

**Recommendation:** Accept with minor revision

**Comment:**

Two reviewers recommend acceptance (although one only leans toward acceptance) while one reviewer leans toward rejection. After examination of the paper, reviews, and responses the action editor sides with acceptance. All reviewers agree that the work is in some dimensions simpler than other approaches, although there is disagreement about the degree and novelty of the proposed architecture and sparse multi-scale attention. Furthermore all reviewers accept the empirical results on accuracy and speed, even if further results are requested, and the provided experiments are interpreted and justified in the discussion. The reviewer ajHh has made a case against the technical significance and novelty of the submission, with detailed discussion, but the action editor overrides this criticism in light of the TMLR criteria. The reviewer cMeH has pointed out concerns about the scope and magnitude of the claims, but these have been discussed in the rebuttal, and can be addressed by minor edits in the revision.

On the balance, the contribution of this work in the investigation of sparse attention for detection merits acceptance. The experiments show the feasibility and efficiency of this choice compared to multi-scale representations and detection architectures and demonstrate generality by addressing object detection, instance segmentation, and panoptic segmentation.

The action editor recommends acceptance with minor revisions in order to incorporate the feedback from review. In particular, the following suggestions deserve to be highlighted:

- Please incorporate the response https://openreview.net/forum?id=6LO1y8ZE0F&noteId=hxjDMmn5S8 to explain the fairness of the comparisons and additional clarifications.
- Please incorporate the response https://openreview.net/forum?id=6LO1y8ZE0F&noteId=r09UbE17jE to address the justified questions by reviewer ajHh. It is important that the controlled comparisons and the analysis of dependence on the backbone and model size are there, but these might need further signposting for the reader to easily find these results.
- Please incorporate the nuanced explanation of their contribution in this final response https://openreview.net/forum?id=6LO1y8ZE0F&noteId=yszK6UrbBy  in the introduction or discussion of the paper. The action editor agrees that the empirical results are non-trivial and the success of the proposed sparse attention across a single feature scale is informative.
- Please make minor edits to revise the claims, as discussed by the authors with cMeH, to propose SIMPLR as a viable alternative rather than proof that multi-scale design is unnecessary.

The action editor thanks the authors and the reviewers for engaging in the TMLR process, and congratulates the authors on their paper!

**Audience:**

In the final reviews and recommendations the two reviewers vote yes on there being an audience while one reviewer votes no. The action editors acknowledges the concern that there could not be an audience, which has been expressed by ajHh due to the amount of technical sophistication and the degree of novelty, but sides with yes. To quote the [TMLR evaluation criteria](https://jmlr.org/tmlr/editorial-policies.html#evaluation), papers should be accepted if they satisfy the claims/evidence and audience questions _even if the contribution or significance of the work is modest_.

The empirical results and engineering contributions of interest to the community:

> shows that a sparse scale-aware attention can be relatively fast and have good performance vs commonly adopted architecture choices (ViTDet, Deformable DETR) [cMeH]
> Gains in speed and accuracy have been reported, e.g. 1.4x faster than ViT-B based Deformable DETR (+1.1 box AP) and BoxeR (+0.3 box AP) baselines [cMeH]
>  it has competitive (even SOTA) box mAP and mask mAP and is competitive with the SOTA in general and is fast and arguably a simpler approach [TLhE]
> motivation behind this work is compelling, as it explores a promising direction focused on developing simple and straightforward architectural designs for fundamental visual recognition tasks, including object detection [ajHh]

Transformers are now the default architecture for detection in computer vision and there has been much work exploring the design of detectors. The focus of this work on scale, and in particular on how to simplify multi-scale representation (= backbone) and inference (= detection head), give it a clear purpose and broad relevance to research on detection. This follows from the popularity of multi-scale architectures.

Taking a step back, this kind of work contributes to making architectures less specific to a given task and more general across tasks. This convergence in methods is a strong current in deep learning for vision, and so simplifications and empirical results in this direction are of interest to the community.

**Claims And Evidence:**

In the final reviews and recommendations the two reviewers vote yes on the agreement of the claims and evidence while one reviewer votes no. The action editor sides with the satisfactory agreement of the claims and evidence save for one point detailed below. Otherwise the points made about the method and the reporting and interpretation of the results are accurate and supported (with fair comparisons, relevant baselines, and qualifications that respect the scope of this work and related work).

However one key point needs addressing, which is the agreement of claims and evidence as emphasized by TMLR. In this case a different perspective would be more measured and informative to the audience w.r.t. the relation between SIMPLR and multi-scale methods:

> A central claim in the paper is that multiscale feature maps and pyramid designs _are unnecessary_, but there is a lack of experimental evidence for this scientific claim. [cMeH]

As submitted, a reasonable reader could see a degree of overclaiming, because the results do not show strict dominance of SIMPLR over hierarchical/multi-scale designs. it is more defensible to claim that SIMPLR is an adequate alternative, as suggested by review, and as discussed and agreed by the authors. Provided this discussion this negative for the claims and evidence has been defused.

---

> ### Author Response · Authors · 2025-02-19
>
> Dear Action Editor,
>
> We have updated the camera-ready version of our paper as requested. Please let us know if any further action is needed.
>
> Best regards,
> Authors